# VISUAL DIFFUSION MODELS ARE GEOMETRIC SOLVERS

## ABSTRACT

In this paper we show that visual diffusion models can serve as effective geometric solvers: they can directly reason about geometric problems by working in pixel space. We first demonstrate this on the Inscribed Square Problem, a long-standing problem in geometry that asks whether every Jordan curve contains four points forming a square. We then extend the approach to two other well-known hard geometric problems: the Steiner Tree Problem and the Simple Polygon Problem. Our method treats each problem instance as an image and trains a standard visual diffusion model that transforms Gaussian noise into an image representing a valid approximate solution that closely matches the exact one. The model learns to transform noisy geometric structures into correct configurations, effectively recasting geometric reasoning as image generation. Unlike prior work that necessitates specialized architectures and domain-specific adaptations when applying diffusion to parametric geometric representations, we employ a standard visual diffusion model that operates on the visual representation of the problem. This simplicity highlights a surprising bridge between generative modeling and geometric problem solving. Beyond the specific problems studied here, our results point toward a broader paradigm: operating in image space provides a general and practical framework for approximating notoriously hard problems, and opens the door to tackling a far wider class of challenging geometric tasks.

## 1 INTRODUCTION

Diffusion models have emerged as a transformative force in generative AI. Initially developed for image synthesis, they have quickly proven to be among the most powerful and versatile generative models across a wide range of media, including audio, video, and 3D content. Their ability to progressively denoise random signals into coherent and high-fidelity samples has enabled breakthrough applications, from photorealistic image generation to controllable editing and cross-modal translation. Beyond their remarkable empirical success, diffusion models are increasingly recognized as a general framework for modeling complex, multimodal distributions.

In this work, we take a different perspective on diffusion models: rather than focusing on their creative generative capacity, we demonstrate their potential as solvers of hard geometric problems. We show that the sampling process of diffusion can be harnessed to directly reason about and discover geometric structures, guided only by pixel-level formulations of the problem. This visual diffusion approach allows us to treat abstract geometric challenges as image generation tasks, bridging the gap between visual synthesis and mathematical problem-solving.

Diffusion models have been used in various contexts to tackle optimization and reasoning problems, including combinatorial tasks such as the traveling salesman problem (Sun & Yang, 2023; Li et al., 2023; Sanokowski et al., 2025). These approaches typically formulate the problem in symbolic or graph-based representations, leveraging the probabilistic nature of diffusion to search solution spaces. In contrast, our method operates purely in the visual domain. By representing geometric problems as images and reasoning directly in pixel space, we exploit the intrinsic strength of diffusion models in handling multimodal distributions and ambiguous solutions. This visual formulation makes our approach fundamentally distinct from prior problem-solving applications of diffusion.

To ground our approach, we begin with the Inscribed Square Problem, a long-standing problem that asks whether every simple closed curve in the plane admits an inscribed square. The problem is still

unsolved in the general case. Furthermore, a given curve may admit multiple and often very different inscribed squares, and enumerating them is non-trivial even in restricted settings (van Heijst, 2014). This multiplicity naturally forms a distribution, which makes the problem especially well suited to diffusion models. Our method addresses it in an unexpected way, operating directly in image space on a visual representation of the geometric challenge. By starting from different random seeds, the model can uncover diverse valid squares, each corresponding to a distinct solution of the problem. Figure 1 illustrates the setting and highlights both the complexity and the variability of possible solutions.

We next illustrate the competence of our diffusion-in-image-space approach on two additional hard geometric problems. The first is the Steiner Tree Problem, which asks for the shortest possible network connecting a given set of points. Its solution may introduce auxiliary nodes, known as Steiner points, and finding the optimal configuration is NP-hard (Garey et al., 1977). The second is the problem of connecting a set of points into a simple polygon of maximum area, which was featured in the CG:SHOP global optimization challenge of 2019 (Demaine et al., 2022). This task is known for its combinatorial complexity and strict geometric constraints, and is also NP-hard. As with the inscribed square problem, our method addresses these problems directly in image space, operating in the pixel domain. While discretization at finite resolution imposes limitations, it nonetheless enables valid approximations to problems that are otherwise extremely hard, with solutions that can be naturally refined. We evaluate this aspect rigorously in the paper.

Training follows a deliberately simple yet effective strategy. We generate a large distribution of valid solutions directly in image space and train a diffusion process to denoise random Gaussian samples into this distribution. This strategy builds

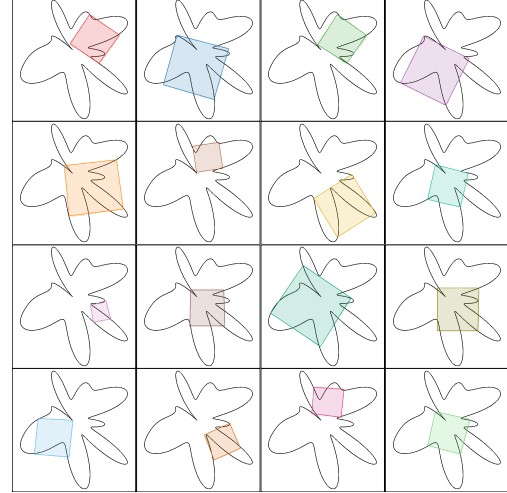

Figure 1: We introduce a visual diffusion approach to solving hard geometric problems directly in pixel space. Shown here on the Inscribed Square Problem, where we task the model with finding a square such that all of its four vertices lie on a given curve. Our method uncovers diverse approximate solutions, corresponding to different random seeds.

on the observation that, in many cases, constructing examples of valid solutions is far easier than deriving one for a specific input instance. Our image-space formulation therefore offers notable simplicity: it requires no elaborate encodings or specialized representations. At the same time, it provides a natural entry point to a wide spectrum of hard geometric problems that admit natural visual representations, beyond the three studied in this work.

## 2 RELATED WORK

Several works have attempted to use diffusion models in order to learn to solve problems for which no known efficient algorithm exists, mostly of combinatorial nature. Most existing diffusion works operate directly on the parameter space of the problem. DIFUSCO (Sun & Yang, 2023), a graph-based diffusion framework, casts a broad family of NP-complete problems into $\{0,1\}^N$ indicator vectors and learns a denoiser over graphs. They compare continuous (Gaussian) and discrete (Bernoulli) noise processes, and show strong results on traveling salesman problem (TSP) and maximum independent set (MIS). Complementing the supervised approach, the T2T (Li et al., 2023) line of work learns a distribution of high-quality solutions during training and then performs gradient-guided optimization at test time by iteratively noising the current solution and denoising while guiding towards lower energy solutions of some relaxed objective, achieving competitive quality–efficiency trade-offs on TSP and MIS. Fast T2T (Li et al., 2024) significantly speeds this up via an optimization-consistency objective, matching or surpassing multi-step diffusion solvers performance with single-step generation plus a single gradient step.

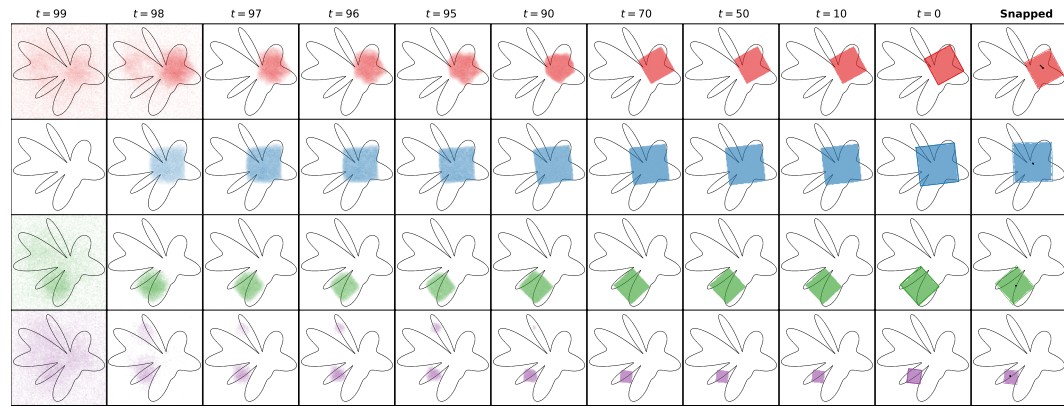

Figure 2: **Inscribed square $x_0$ predictions across denoising steps.** Each row corresponds to a different seed (inscribed square). Columns show selected $x_0$ predictions for decreasing timesteps $t$ from left to right (leftmost: $t=T$; penultimate: $t=0$). For $t \neq 0$ we render only the filled mask; at $t=0$ we also draw square edges and the minimum-area bounding box. The rightmost column ("snapped") shows the rigidly snapped version of the $t=0$ prediction on the curve, with an arrow from the original centroid to the snapped centroid.

Closer to our approach, some methods solve constrained problems in pixel-space representation. Graikos et al. (2023) train an unconditional diffusion model on pixel-space representations of TSP instances. They then solve new instances with stochastic optimization using a differential renderer, utilizing the prior of the learned model. Differently from us, they optimize a parametric representation of the solution to a given instance, while we generate solutions with DDIM sampling of a conditional model. Wewer et al. (2025) train a noise-prediction UNet in pixel space on a visual representation of Sudoku, which is another NP-hard combinatorial problem. They depart from fully-parallel image diffusion by (i) assigning individual noise levels to patches, and (ii) sampling patches in a learned order or a hand-crafted order. They demonstrate that sampling order matters, and can substantially outperform a conditional DDPM baseline that operates fully in parallel.

## 3  INSCRIBED SQUARE PROBLEM

We present our visual diffusion approach as a solver for hard geometric problems by structuring the paper around a set of case studies on well-known challenges. Each case study begins with the problem statement and its mathematical context, followed by a brief review of existing methods. We then describe how our image-space diffusion formulation addresses the task, highlighting both its capabilities and limitations. The first and central case we study is the inscribed square problem, which serves as an illustrative entry point into our approach.

**Problem Statement**  The *Inscribed Square Problem*, also known as Toeplitz's *Square Peg Problem* first posed in 1911, asks whether every Jordan curve in the Euclidean plane contains four points that form a square. Formally, the conjecture states that for every Jordan curve $C \subset \mathbb{R}^2$, there exist four points $\{p_1, p_2, p_3, p_4\} \subset C$ such that $p_1, p_2, p_3, p_4$ are the vertices of a non-degenerate square.

Figure 3 illustrates this setting, showing a curve with several inscribed squares.

The problem has since been resolved in several restricted settings. Some works show the conjecture holds for convex and piecewise analytic curves (Emch, 1913; 1915; 1916) and later results prove it for $C^1$-smooth curves and for curves of finite total curvature or generic $C^1$ curves (Stromquist, 1989; Cantarella et al., 2021). More

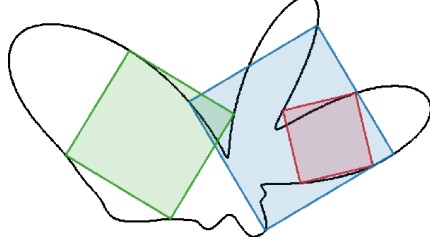

Figure 3: Example of a curve (black) with three inscribed squares. Note that the inscribed squares are defined only by having all four vertices on the curve: they need not be fully contained within the curve, and they may overlap with each other.

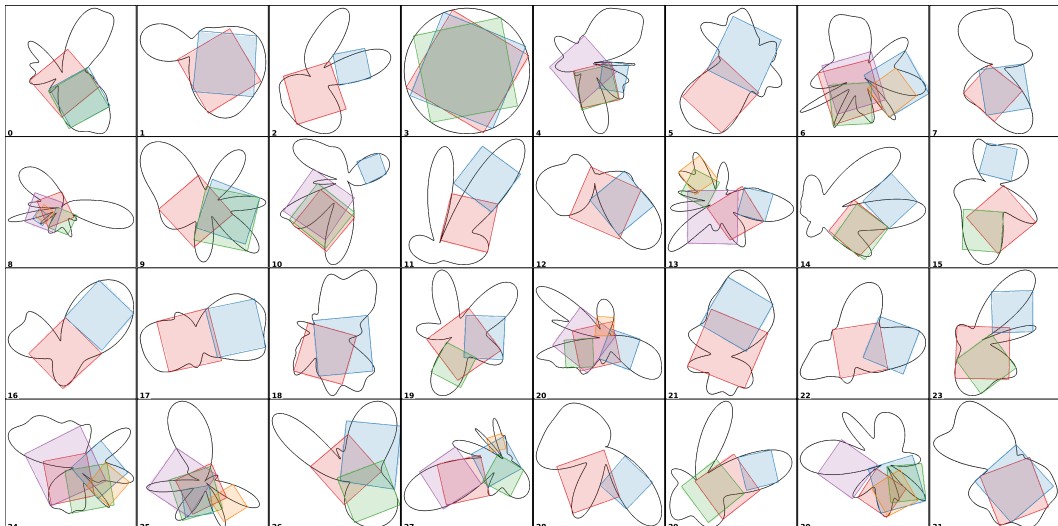

Figure 4: Solutions produced by our model. Each Jordan curve (black) is accompanied by predicted inscribed squares (colored).

recent work demonstrates validity under additional low-regularity assumptions (Tao, 2017), yet it remains open for the general Jordan curve case.

**Existing Methods**   Algorithmic approaches for finding inscribed squares exist primarily for discrete or polygonal cases. For convex polygons, efficient procedures have been developed to detect or enumerate inscribed squares in subquadratic time (Chazelle, 1983; Sharir & Toledo, 1994). Discrete relaxations on grid polygons have likewise been studied and verified computationally for bounded cases (Pettersson et al., 2014). All these methods represent the curve symbolically or combinatorially, relying on exact geometric descriptions. In contrast, our work formulates the problem directly in pixel space.

**Method**   We address the inscribed square problem by formulating it entirely in image space and training a conditional diffusion model to recover inscribed squares from noisy inputs. To this end, we generated a dataset of synthetic samples. Each sample consists of a smooth Jordan curve $C$ together with one to five inscribed squares. The curves were constructed procedurally such that they pass through the square vertices while avoiding self-intersections, and both curves and squares were rasterized into $128 \times 128$ binary images. Each training example pairs a curve image with one inscribed square. Full details of the curve-generation process are provided in the supplementary material, under Appendix B.1.

Our model follows the standard diffusion framework with a U-Net (Ronneberger et al., 2015) backbone with self-attention layers (Vaswani et al., 2017), similar to those employed in text-to-image diffusion models such as Stable Diffusion (Rombach et al., 2022). The conditioning signal is the clean binary image of the curve, while the ground truth $x_0$ is the clean image of the square, both represented in two-dimensional pixel space. The conditioning image is concatenated as an additional channel to the noisy input $x_t$ at each timestep, effectively treating the curve as a non-noisy color channel. We train with 100 denoising steps using a standard noise schedule and mean-squared error objective, enabling the model to transform noisy square images into valid inscribed squares consistent with the given curve.

During sampling, at each step the network predicts an $x_0$ estimate of the square conditioned on the curve. To inspect this behavior, we visualize $x_0$ predictions at a subset of timesteps arranged left-to-right from $t{=}T$ to $t{=}0$ (penultimate column) in Fig. 2.

**Square Enhancement (Snapping)**   As a final post-processing step, we refine each predicted square $\widehat{S}$ by snapping it to the conditioning curve $C$ of the respective problem instance. Let $V(S) = \{p_1, p_2, p_3, p_4\}$ denote the set of four vertices of a square $S$. To quantify how well $S$ aligns with $C$, we define the negative average corner-to-curve distance as the alignment score:

Table 1: **Evaluation Results.** We report alignment $\mathcal{A}(S, C)$ and squareness $\mathcal{Q}$ under three conditions: before snapping, after snapping, and ground truth (GT).

| w/o snapping | | w/ snapping | | Data (GT) | |
|---|---|---|---|---|---|
| Align ↑ | Square ↑ | Align ↑ | Square ↑ | Align ↑ | Square ↑ |
| -1.60 | 0.892 | -0.90 | 0.891 | -0.14 | 0.924 |

$$\mathcal{A}(S, C) \ = \ -\frac{1}{4} \sum_{p \in V(S)} \mathrm{dist}(p, C), \tag{1}$$

where $\mathrm{dist}(p, C)$ is the Euclidean distance from vertex $p$ to the curve $C$.

We then apply a small rigid transformation $(R_\theta, t)$ to $\widehat{S}$ and select the configuration that maximizes this alignment score. In practice, we approximate this optimization with a coarse grid search over small rotations and translations (see Appendix B.2 for details). This procedure nudges the predicted square so that its corners sit more closely on $C$, as visualized in the rightmost column of Fig. 2.

**Evaluation**   Since multiple valid squares could potentially fit the curves beyond the ground truth squares used in construction, we avoid comparing distances to the closest ground truth square and instead focus on evaluating the geometric properties of our generated solutions.

We evaluate our method along two complementary axes: *alignment* and *quality*. For alignment, we report the score $\mathcal{A}(S, C)$ defined in Eq. 1, which directly measures how well the vertices of a predicted square lie on the conditioning curve. For quality, we introduce a *squareness metric* that captures how close a predicted shape is to a valid square. Given a predicted square $S$, let $\mathrm{area}(S)$ denote its contour area, and let $(w, h)$ denote the side lengths of its minimum-area enclosing rectangle. We define:

$$\mathcal{Q}(S) \ = \ \frac{\mathrm{area}(S)}{w \cdot h} \ \cdot \ \exp\!\Big(- 2 \left| \tfrac{\max(w,h)}{\min(w,h)} - 1 \right|\Big). \tag{2}$$

This produces a score in $[0, 1]$ that is high only when $S$ tightly fills a nearly equilateral rectangle, i.e., when it closely resembles a true square.

We report both alignment and quality metrics under three conditions: (i) predictions before snapping, (ii) predictions after snapping, and (iii) the ground-truth squares from the dataset (Tab. 1). This evaluation disentangles the intrinsic generative ability of the diffusion model from the gains achieved by the geometric snapping refinement.

**Interpretation of Evaluation Results**   The evaluation demonstrates that our model consistently produces shapes that closely approximate true inscribed squares with strong accuracy. Even though the alignment of predicted squares with the conditioning curve is not always pixel-perfect, the snapping step leads to a substantial refinement, bringing the results impressively close to the ground truth. In practice, this shows that the diffusion process is highly effective at capturing both the structural regularity and the correct placement of squares, with only minimal residual deviation that often manifests at the sub-pixel level. Importantly, such deviations are expected given the inherent discretization of the pixel domain, which naturally puts an upper bound even on the ground truth results. Within this setting, our method reliably recovers high-quality approximations of valid inscribed squares. The reported numbers confirm that the model does not merely suggest plausible candidates but in fact achieves precise and robust approximations, validating the strength of the visual diffusion framework as a solver for this classical geometric challenge.

## 4   STEINER TREE PROBLEM

The second problem we cover in our case study is that of the Steiner Tree Problem. The Steiner tree problem asks, given a set of terminal points, to find a network of minimum total length that connects all terminals, where the construction is allowed to introduce additional points (Steiner points) to reduce length. In the Euclidean variant, which we focus on, Steiner points may be placed anywhere in the plane. The Steiner formulation is central to many applications where minimizing connection cost is critical. Some typical use cases for it include telecommunication (Voss, 2006), PCB routing (Chu & Wong, 2008), as well as infrastructure layout (roads, pipelines) (Schwartz & Stückelberger, 2008; Cui et al., 2021).

**Problem statement.** Formally, given a finite set of terminals $P = \{p_1, \ldots, p_n\} \subset \mathbb{R}^2$, the Euclidean Steiner Tree (EST) problem asks for a straight-line embedded tree $T = (V, E)$ with $P \subseteq V$ that minimizes total Euclidean length $L(T) = \sum_{uv \in E} \|u - v\|_2$. Vertices in $V \setminus P$ are *Steiner points*. An optimal solution is called a *Steiner minimal tree* (SMT) and its length is denoted $L^\star(P)$ (Gilbert & Pollak, 1968). Figure 5 illustrates an instance of the problem and its optimal solution.

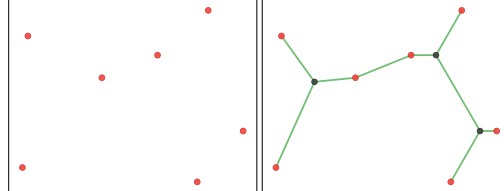

Figure 5: Example of a Steiner Minimal Tree. Left: The input terminal nodes, colored in red. Right: The Steiner Minimal Tree for this instance, where auxiliary Steiner points are colored in dark gray.

**Complexity and Approximability.** For general $n$ (the number of initial given points), the EST problem is NP-hard already in the plane (Garey et al., 1977). However, there is a polynomial-time approximation scheme (PTAS) for Euclidean Steiner tree in fixed dimensions: for any fixed $\varepsilon > 0$ one can compute a $(1+\varepsilon)$-approximate tree in time polynomial in $n$ (for fixed $\varepsilon$ and dimension).

**Structure and Properties** SMTs in the plane are highly structured: (i) no two edges cross; (ii) each Steiner point has degree exactly 3 and the three incident edges meet at $120°$; (iii) all angles in the tree are at least $120°$ (at terminals that have degree $> 1$); (iv) the number of Steiner points is at most $n - 2$; and (v) all Steiner points lie in the convex hull of $P$ (Hwang & Richards, 1992; Brazil et al., 2004). These properties provide strong geometric constraints that are exploited by both exact and approximate methods, as discussed next.

**Algorithmic Approaches.** Exact algorithms typically rely on generating locally optimal full Steiner trees and then selecting a minimum-length subset, with implementations such as GEOSTEINER solving large 2D instances (Warme et al., 2000; Juhl et al., 2018). On the approximation side, PTASes based on dissection or guillotine methods provide near-optimal guarantees in fixed dimensions (Arora, 1998; Mitchell, 1999).

**Learning-based Approaches** While classical exact/approximate algorithms dominate practice, there have been several attempts to use learning-based solvers for the problem. For the Euclidean case, Wang et al. propose *Deep-Steiner*, which casts SMT construction as a sequential decision process: it discretizes the continuous search space, prunes candidates via KNN/MST neighborhoods, and then uses an attention-based policy trained with REINFORCE to add Steiner points iteratively (Wang et al., 2022). For non-Euclidean variants (rectilinear SMT and graph STP), several learning-based methods have been explored, including RL and GNN-driven solvers and mixed neural–algorithmic pipelines (Liu et al., 2021; 2024; Kahng et al., 2023; Ahmed et al., 2021; Du et al., 2021; Park et al., 2025).

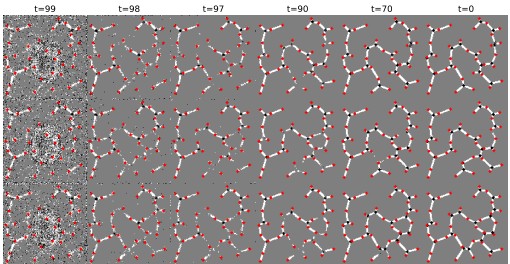

Figure 6: **Steiner tree $x_0$ predictions across denoising steps.** Each row corresponds to a different seed. Columns show selected $x_0$ predictions for decreasing timesteps $t$ from left to right (leftmost: $t=T$; rightmost: $t=0$). Input points are overlaid in red.

**Method** We generate a synthetic dataset by sampling random points (between 10 and 20 for each instance), and finding their SMT using the GeoSteiner solver (Juhl et al., 2018). Each solution is then rasterized into a grayscale image with different values for edges, nodes and background. The full details for data generation are found in Appendix B.3. We employ the same U-Net backbone as in the inscribed square experiment. The conditioning input consists of the rasterized terminal points, concatenated as an additional channel to the noisy input $x_t$ at each denoising step.

Recovering a graph structure from the generated image is performed in two stages. We begin by node detection, where we binarize the output with a threshold and detect centers of connected components. The centroid of each blob is then taken as a node position, with nodes falling within a small radius of an input terminal being snapped to that terminal's location. In the second stage, we extract edges by considering the complete graph over the detected nodes. For each candidate edge,

Table 2: **Steiner Tree Evaluation Results.** Comparison of our method, MST, and random solutions. Reported are valid tree rates and mean Euclidean length ratios ($\pm$ std) relative to the optimal solution across input point ranges.

| Number of Input Points | Valid Trees Rate | Ours Ratio Mean $\pm$ Std | MST Ratio Mean $\pm$ Std | Random Ratio Mean $\pm$ Std |
|---|---|---|---|---|
| 10-20 | 0.996 | $1.0008 \pm 0.0005$ | $1.0363 \pm 0.0124$ | $1.8344 \pm 0.2363$ |
| 21-30 | 0.986 | $1.0018 \pm 0.0011$ | $1.0416 \pm 0.0095$ | $1.9044 \pm 0.1827$ |
| 31-40 | 0.834 | $1.0044 \pm 0.0035$ | $1.0470 \pm 0.0079$ | $1.8981 \pm 0.1656$ |
| 41-50 | 0.334 | $1.0092 \pm 0.0055$ | $1.0522 \pm 0.0072$ | $1.8605 \pm 0.1425$ |

Figure 7: Optimal solutions (left) vs. our model's solutions (middle) and the difference between them (right). Input points are overlaid over both optimal and produced solutions as red circles.

we compute the fraction of pixels along the straight line segment that are marked as foreground. If this fraction exceeds a threshold (70% in our implementation), the edge is retained. If two vertices are very close to each other, we assume they are connected via an edge. In cases of ambiguity where multiple potential edges with a shared node overlap, we retain the shortest one and discard the rest.

**Evaluation** We evaluate our trained model on a test set containing instances with 10-20 input points, matching the number of points seen during training, and four other test sets containing 11-20, 21-30, 31-40 and 41-50 input points. After extracting the graph from the generated solution, we check the validity of the solution by verifying that the resulting graph is a tree and that it contains all of the input points . If the solution is valid, we then measure the total Euclidean length of the tree. For each instance, we generate in parallel 10 solutions from different noise seeds and select the one with minimal total edge length that is also valid. In Table 2 we report for each test set the rate of valid solutions as well as the mean ratio between the total Euclidean length of the best solution produced by our model compared to that of the optimal solution ($L^\star(P)$). For comparison, we also report the ratio between the total Euclidean length of a random planar tree and the optimal solution and that of the solution produced by the minimum spanning tree of the full graph and $L^\star(P)$.

Our model is able to successfully produce high quality solutions even for instances with markedly more input points than were seen during training, and often produces solutions that align with the optimal ones (see Figure 7).

While there is generally a one-to-one coupling between an input and the optimal solution, for some instances the model still produces variations, often of similar quality, for different noise initializations (see Figure 6). However, some of these variations can happen to be invalid, especially for instances with a large number of input points. This is evident in the third row, where the solution produced by the noise initialization contains a loop and is not a tree.

## 5 MAXIMUM AREA POLYGON PROBLEM

The third problem we attempt to tackle with our approach is the Maximum Area Polygonization Problem (MAXAP), a well-established problem in computational geometry. Given a set of vertices in the plane, the problem asks to find a simple polygon (a polygon that does not intersect itself and has no holes) that passes through all the vertices and has the largest possible area.

MAXAP is known to be NP-complete (Fekete, 1992; Fekete & Pulleyblank, 1993) and difficult to solve both in theory and in practice (Fekete et al., 2021), with no known algorithm that provides

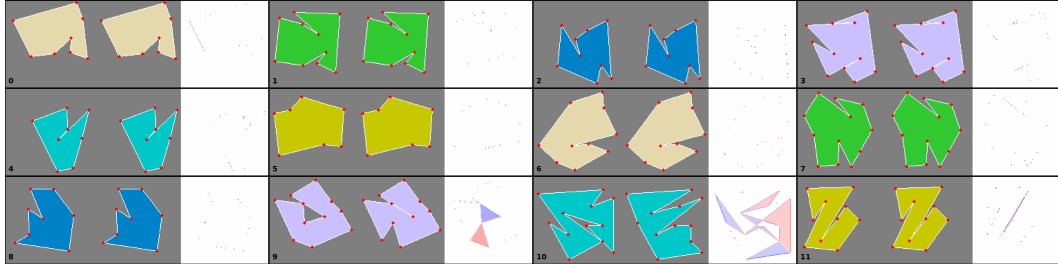

Figure 8: Qualitative examples of maximum area polygons (left) vs. polygons produced by our model (middle) and the difference between them (right). Areas depicted in red and blue in the difference map correspond to regions that are exclusive to the optimal solution and our solution, respectively. It can be noticed that even in cases where there is a disparity between the optimal solution and the one produced by the model, the area difference between the exclusive regions tends to be small, amounting to a solution of similar quality. Input points are overlaid over both optimal and produced solutions as red circles.

Table 3: **Maximum Area Polygon Evaluation Results.** Comparison of our method, random polygons, and optimal solutions. Metrics include polygon validity rate, mean area ratio ($\pm$ std), and optimal solution rate for different input point ranges.

| Number of Input Points | Valid Polygons Rate | Ours Ratio Mean $\pm$ Std | Random Polygon Ratio Mean $\pm$ Std | Optimal Solutions Rate |
|---|---|---|---|---|
| 7-12 | 0.953 | $0.9887 \pm 0.0205$ | $0.7711 \pm 0.1361$ | 0.574 |
| 13-15 | 0.620 | $0.9624 \pm 0.0418$ | $0.4779 \pm 0.2717$ | 0.062 |

better than $\frac{1}{2}$-approximation factor in polynomial time. Furthermore, deciding whether there exists a simple polygon that contains strictly more than 2/3 of the area of the convex hull is also NP-complete (Fekete, 1992). Exact approaches based on integer programming are able to solve instances with up to 25 points (Fekete et al., 2021), while a recent mixed-integer-programming approach is able to solve instances with up to 50 points (Hernández-Pérez et al., 2025). Heuristic approaches are commonly applied to larger instances, including constrained triangulations (Lepagnot et al., 2023), simulated annealing (Lepagnot et al., 2023; Goren et al., 2022), and divide-and-conquer strategies for very large point sets of up to 1,000,000 points (Crombez et al., 2022; Goren et al., 2022).

**Method**  To address the maximum area polygonization problem, we adopt the same visual diffusion architecture as in the previous two tasks, using an identical U-Net backbone. As with the previous methods, we generate a synthetic dataset of examples to train the model on. For each training example we sample input points randomly on the grid, and compute their optimal polygonizations through exhaustive search over all valid simple polygons, which is feasible at this scale using a DFS procedure. Each polygon is rasterized to an image, while the input points are rasterized into a separate image that is concatenated as an additional conditioning channel. The data generation procedure is described in full in Appendix B.4

At inference time, we recover the polygon structure from the generated image by testing candidate edges between all point pairs. Each edge is retained if more than 70% of its pixels align with foreground edge pixels in the output. The resulting set of edges is then validated to ensure that no intersections occur, with mild tolerance for nearly parallel overlaps. Finally, we search for a simple cycle that passes through all input vertices, which we consider the recovered polygonization produced by our model.

**Evaluation**  We evaluate the trained model on a test set containing 7–12 points, matching the range used during training. To further assess generalization, we also test the model on a set with 13–15 points . For each instance, we generate 10 candidate solutions from different random seeds and select the one that achieves the largest area while remaining valid. In Table 3, we report the mean and standard deviation of the ratio between the best solution found for each instance and the corresponding optimal solution, and show a comparison against random simple polygons on the same point sets. We also report the rate of valid polygons (the proportion of instances for which a valid polygon was produced), as well as the frequency with which the recovered polygon coincides exactly with the optimal one.

Figure 8 shows qualitative examples of polygons produced by our model. In many cases, the generated polygons align almost perfectly with the optimal ones. When discrepancies occur, the differences often balance out, with areas lost in one region largely compensated elsewhere (see instances 9, 10 in the figure). Owing to the non-local nature of the problem, instances with a larger number of points are substantially more challenging, and the rate of valid solutions drops noticeably for the 13–15 point test set. Typical failure cases include polygons that do not pass through all input points or that contain holes (see the third and fourth rows in Figure 9). Nevertheless, whenever a valid polygon is produced, its area is typically very close to that of the optimal solution.

We note that for this problem, like the last one, there is generally only a single optimal solution per problem instance. Therefore the benefit of training conditional diffusion models which are generally used to learn conditional distributions in order to solve these instances is not immediately clear. In Appendix A we demonstrate on the MAXAP problem that there is a performance advantage over using a regression model even for problems of this kind.

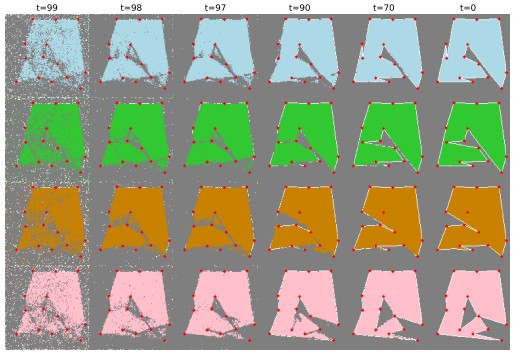

Figure 9: **Maximum area polygon $x_0$ predictions across denoising steps.** Each row corresponds to a different seed. Columns show selected $x_0$ predictions for decreasing timesteps $t$ from left to right (leftmost: $t=T$; rightmost: $t=0$). Input points are overlayed in red.

## 6 Discussion and Conclusions

In this work, we presented visual diffusion as a general framework for approximating solutions to notoriously hard geometric problems. Through three case studies, the *Inscribed Square* problem, the *Steiner Tree* Problem, and the *Simple Polygon* Problem, we demonstrated that diffusion models can operate in image space to uncover valid geometric structures.

We do not claim that our method outperforms specialized solvers tailored to any single problem. Indeed, for each of these problems, carefully designed algorithms may yield more efficient or more accurate solutions. Instead, our contribution is to reveal a paradigm: visual diffusion provides a single, simple framework that applies across a diverse set of problems without requiring custom formulations. Specifically, each task uses the very same diffusion architecture without modification, varying only in task-specific training data. Our approach produces accurate and diverse approximations, naturally recovering multiple valid solutions through diffusion, as illustrated in the Inscribed Square problem. These solutions can be further refined if desired. Importantly, we also observe that models trained on relatively simple instances generalize to more complex inputs, such as handling a larger number of points than those seen in training. This behavior is particularly valuable for problems where complexity grows with the number of points. This contrasts with traditional geometric solvers, whose runtime typically grows polynomially or even exponentially with input size.

Despite the diversity of the problems they are trained to solve, the models exhibit a consistent behavior, evident in the denoising progression (Figures 2, 6 and 9). Already in the early steps of the sampling process, the global structure of the solution becomes apparent, suggesting that the essence of the solution lies primarily in low-frequency geometric features that can be recovered quickly. The subsequent denoising steps refine these structures to achieve high accuracy. This observation indicates that inference time could be further optimized, with only a small trade-off in precision, by using denoising schedulers that allocate more of the sampling steps budget to earlier timesteps. More broadly, this progressive reasoning mirrors how humans intuitively reason about geometric problems: they first sketch a mental image of a coarse solution in their mind, which they can then try to translate to a concrete solution, with details settled upon after the initial structure is clear.

The key message of this work is that image diffusion models, long celebrated for their generative capacity, also serve as geometric solvers. This outlook opens the door to exploring a wide spectrum of geometric challenges under a single methodology, and suggests new opportunities for bridging generative modeling with geometric problem solving.

ETHICS STATEMENT

Our experiments were conducted using 4 GTX 3090 GPUs on a shared GPU cluster, with training times of around 20 hours per geometric problem. While we acknowledge the environmental implications of any computational research, our resource requirements are modest compared to many modern deep learning projects. To further minimize environmental impact, we utilized shared institutional resources that allow for efficient GPU utilization across multiple research projects, and we will release all pre-trained models to prevent redundant retraining. Researchers can reproduce our results or build upon our work using the released checkpoints without incurring additional training costs.

REPRODUCIBILITY STATEMENT

In Appendix B.5 and Appendix B.6 we provide details about the hyper-parameters we used the model and the training runs. We will also release code and training data for all three problems, allowing for convenient reproduction of our reported experimental results.

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

# APPENDIX

## A   REGRESSION MODEL ABLATION

For certain geometric problems, each input instance admits a unique optimal solution. In such cases, one may wonder whether training a diffusion model is necessary, since the task appears to reduce to learning a deterministic mapping. Indeed, the conditional distribution to be learned degenerates into a collection of Dirac delta functions.

To examine this, we compared our diffusion framework with a direct regression baseline. For the Maximum Area Polygonization problem, we trained a regression model with the same U-Net backbone as our diffusion model, using identical training data and budget. The regression model outputs a polygon image in a single forward pass, conditioned only on the rasterized input points.

While the regression model succeeds on simpler instances, it often produces polygons with blurry edges or missing segments for more complex cases. This degrades our polygon extraction stage, which frequently fails to recover a valid polygon from such outputs. The quantitative results, shown in the "Regression Valid Polygons Rate" column of Table 4, confirm this limitation.

In contrast, the diffusion model retains a key advantage: stochasticity. By conditioning on both the input points and a noise vector, we can generate multiple candidate solutions and resample until a valid polygon is obtained. This property directly improves the robustness of the approach, and highlights why diffusion remains beneficial even in problems that might superficially appear deterministic. This ablation study demonstrates our key finding: diffusion models offer a versatile and robust solution framework. This framework can also surpass the capabilities of deterministic regression approaches, even in scenarios where the underlying function maintains a one-to-one correspondence.

Table 4: **Regression Model Evaluation Results.** Comparison of diffusion (best-of-10) and regression models. Reported are valid polygon rates and mean area ratios ($\pm$ std) across input point ranges.

| Number of Input Points | Diffusion Valid Polygons Rate | Regression Valid Polygons Rate | Diffusion Ratio Mean $\pm$ Std | Regression Ratio Mean $\pm$ Std |
|---|---|---|---|---|
| 7-12 | 0.953 | 0.361 | $0.9887 \pm 0.0205$ | $0.9994 \pm 0.0025$ |
| 13-15 | 0.620 | 0.016 | $0.9624 \pm 0.0418$ | $0.9988 \pm 0.0031$ |

## B   IMPLEMENTATION DETAILS

### B.1   CURVE GENERATION

For harmonic-based curves, we construct a random radial profile

$$r(\theta) = 1 + \sum_{h=1}^{H} \rho_h \sin(h\theta + \phi_h),$$

with $H$ sampled uniformly from $[H_{\min}, H_{\max}]$ and amplitudes $\rho_h$ drawn from a decaying envelope to produce smooth perturbations. Square vertices are first placed in Cartesian coordinates, then converted to polar coordinates $(\theta_i, r_i)$. A periodic cubic spline is fit to the radius corrections $r_i - r(\theta_i)$, ensuring that the resulting contour passes exactly through all square vertices. To guarantee validity, we enforce periodicity in the spline domain and regenerate until a non-self-intersecting (Jordan) curve is obtained. A random global translation is then applied, and both the curve and its inscribed squares are normalized to fit inside $[-1, 1]^2$ before rasterization.

In practice, we used the following parameter ranges: $H \in [6, 30]$, 500 angular samples, square side lengths sampled from $[0.3, 0.7]$, rotations from $[0, 2\pi]$, and global translations up to $0.5$ units. Each curve contains between 1 and 5 inscribed squares.

As an additional augmentation, with probability $0.1$ we replace the harmonic-based curve with a perfect circle of random radius. Since a circle trivially admits infinitely many inscribed squares (obtained by rotation), we sample a small number of representative ones to enrich the dataset. The samples are finally rasterized into $128 \times 128$ binary images, with curves as one-pixel-wide strokes and squares as filled shapes. In total, the dataset contains 100,000 examples.

## B.2 SQUARE ENHANCEMENT

To extract the initial square $\widehat{S}$ from the predicted mask, we fit a contour-aligned minimum-area rectangle and take its four vertices as $V(\widehat{S}) = \{p_i\}_{i=1}^4$. For a candidate rigid transform $(R_\theta, t)$ with $\theta$ in radians and $t \in \mathbb{R}^2$, we form the transformed square

$$S(\theta, t) = R_\theta \widehat{S} + t,$$

with vertices $V(S(\theta, t)) = \{q_i(\theta, t)\}_{i=1}^4$, where $q_i(\theta, t) = R_\theta p_i + t$. We then evaluate the alignment score with the curve $C$ as defined in Eq. 1.

The final snapped square is obtained by selecting the rigid transform that maximizes this score:

$$(\theta^*, t^*) = \arg\max_{\theta, t} \mathcal{A}(S(\theta, t), C), \qquad S^* = R_{\theta^*} \widehat{S} + t^*.$$

We approximate this maximization with a discrete grid search. Specifically, we sample $\theta \in [\theta_{\min}, \theta_{\max}]$ with step $\Delta\theta$, and translations $t = (\Delta x, \Delta y)$ with $\Delta x, \Delta y \in \{-T, \dots, T\}$ in steps of one pixel. For each candidate $(\theta, t)$, the square mask is rigidly warped, its corners recomputed, and $\mathcal{A}(S(\theta, t), C)$ evaluated.

## B.3 STEINER TREE GENERATION

Generation of a synthetic instance begins by sampling $n$ terminal nodes within the unit square, with $n$ drawn uniformly from $[10, 20]$. To prevent rasterization artifacts, we enforce a minimum separation between terminals with rejection sampling. For each sampled configuration, we compute the Steiner Minimal Tree using the GeoSteiner solver (Juhl et al., 2018). The resulting solutions are rasterized into grayscale images of fixed resolution ($128 \times 128$), where terminals and Steiner points are depicted as small filled black circles with a radius of 2 pixels and edges as thin white lines that are 2 pixels wide, while the background is gray. The final dataset contains 1,000,000 instances.

## B.4 MAXIMUM AREA POLYGON GENERATION

For MAXAP instance generation, we first sample $n$ points within the unit square, with $n$ drawn uniformly from $[7, 12]$. Also here we enforce a minimum separation between points with rejection sampling. For each set of point, we exhaustively go over all valid polygon configurations and find the one with the largest area. We employ a backtracking depth-first search to systematically explore all valid simple polygons formed by a given point set. The method fixes an anchor point at the bottommost-leftmost position to eliminate rotational symmetry, then incrementally constructs polygons by selecting vertices in angular order around the centroid. At each step, the algorithm prunes invalid branches by rejecting vertices that would create edge intersections with the existing partial polygon. When a complete polygon is formed, the closing edge is validated for intersections, and the polygon area is computed using the shoelace formula (O'Rourke, 1998). The search maintains the globally optimal solution by comparing areas and updating the best configuration found. This approach guarantees finding the maximum area simple polygon while significantly reducing the exponential search space through geometric pruning, making the $O(n!)$ worst-case complexity manageable and runtime that is fast in practice for small point sets ($n \leq 15$). Finally, the polygon is rasterized into grayscale images of fixed resolution ($128 \times 128$), where the polygon edges are drawn as white lines that are 1 pixel wide, the polygon interior is black and the background is gray. In total the dataset contains 1,000,000 instances.

## B.5 MODEL ARCHITECTURE

Our approach employs a conditional diffusion model based on a U-Net architecture for generating geometric solutions. The U-Net consists of 4 encoder and decoder levels with a base channel count

of 64, following a standard channel progression of $64 \rightarrow 128 \rightarrow 256 \rightarrow 512$ in the encoder path. The model takes a 2-channel input (noisy target and condition images) and produces a single-channel denoised prediction.

Multi-head self-attention with 8 heads is integrated at the bottleneck and at encoder/decoder levels 2 and 3. The attention mechanism uses GroupNorm with 32 groups. Each level incorporates residual blocks with two $3 \times 3$ convolutional layers, BatchNorm, and ReLU activations, along with time embedding injection through learned linear projections. Sinusoidal time embeddings with 128 dimensions condition the model on the diffusion timestep.

### B.6 TRAINING PROCEDURE

The model is trained using the DDIM (Denoising Diffusion Implicit Models) framework with 100 diffusion steps, a linear beta schedule and deterministic sampling ($\eta = 0.0$). We employ an $L_2$ loss on noise prediction, where the model learns to predict the noise $\epsilon$ added to the clean image at timestep $t$:

$$\mathcal{L} = \mathbb{E}_{t,\epsilon \sim \mathcal{N}(0,I)} \left[ \|\epsilon - \epsilon_\theta(x_t, t, c)\|_2^2 \right]. \tag{3}$$

Training is performed using the AdamW optimizer with a learning rate of $6 \times 10^{-4}$ and cosine annealing with warm restarts over 0.5 cycles, including 100 warm-up steps and a minimum learning rate factor of 0.1. We use gradient accumulation over 8 steps with gradient clipping at a maximum norm of 1.0, and a batch size of 128 per GPU. The training employs mixed precision with bfloat16 autocast for efficiency and is distributed across 4 NVIDIA GTX 3090 GPUs for 100 epochs.

## C  LLM USE DISCLOSURE

Large language models (LLMs) were used as part of this paper's writing for the purpose of assistance with editorial refinement as well as literature discovery in certain areas. All content has been carefully reviewed and verified by the authors, and we take full responsibility for the accuracy of the presented research.

