# OpenReview forum: "Visual Diffusion Models are Geometric Solvers"
_ICLR.cc/2026/Conference — ICLR 2026 Conference Withdrawn Submission_

### Official Review · Reviewer_L3Qn · 2025-10-25

**Soundness:** 3
**Presentation:** 3
**Contribution:** 3
**Rating:** 6
**Confidence:** 2

**Summary:**

This paper presents a novel pipeline that extends visual diffusion models to tackle complex geometric problems. The proposed approach reformulates geometric problems as image representations and leverages visual diffusion models to generate approximate solutions. The authors explore three representative geometric problems—the Inscribed Square Problem, the Steiner Tree Problem, and the Simple Polygon Problem—and demonstrate interesting and promising results. Overall, the paper offers an innovative perspective, suggesting that generative models can be applied beyond traditional content generation to address mathematical and geometric problem-solving.

**Strengths:**

1. The idea of applying visual diffusion models to solve geometric problems is both novel and inspiring. It opens a new direction for researchers to explore the potential of generative models not only in creative content generation but also in mathematical problem solving.
2. The paper presents a clear and effective method to transform geometric problems into visual generation tasks, achieving promising results.

**Weaknesses:**

1. While the paper defines several geometric problems, I am concerned about the inherent inaccuracy of representing geometric structures in pixel space. Geometric problems are typically defined with strict mathematical precision, whereas image-based representations can introduce approximation errors. Moreover, many geometric problems—especially optimization problems—require strict constraint satisfaction. Since diffusion models are inherently stochastic, it would be helpful if the authors could further elaborate on how their approach ensures that generated solutions adhere to such constraints.
2. The study is limited to three geometric problems (Inscribed Square, Steiner Tree, and Simple Polygon). The authors could strengthen the discussion by analyzing which types of geometric problems are suitable for diffusion-based approaches and which are not. Such an analysis would provide valuable insight into the applicability boundaries of the proposed method and guide future research on selecting appropriate tools for different problem classes.

**Questions:**

Please refer to weaknesses.

---

### Official Review · Reviewer_KQVd · 2025-10-29

**Soundness:** 3
**Presentation:** 3
**Contribution:** 2
**Rating:** 4
**Confidence:** 3

**Summary:**

This paper demonstrates that standard visual diffusion models can work as effective geometric solvers. The method frames geometric challenges as an image-generation task, training a conditional U-Net to transform Gaussian noise into a pixel-space image of a valid solution, conditioned on a visual representation of the problem (e.g., a curve or a set of points). The framework is evaluated on three problems: the Inscribed Square Problem, the Steiner Tree Problem, and the Maximum Area Polygon Problem. While limited to small-scale instances by its pixel-based nature, the model produces high-quality approximate solutions and can naturally uncover diverse valid solutions for multi-modal problems.

**Strengths:**

- The paper introduces a novel and interesting paradigm by formulating geometric challenges as image generation tasks. This approach is naturally suited to problems with multiple valid solutions, such as the Inscribed Square Problem, as it can uncover diverse answers by sampling from different initial noises.

- The same pipeline was successfully applied to three different and notoriously hard geometric problems. The model produced high-quality solutions for small-scale instances of these problems, demonstrating the viability of using image diffusion models as general geometric solvers.

**Weaknesses:**

- The most critical weakness of the proposed method is the restrictions imposed by image resolution. Solving geometric problems in pixel space fundamentally limits the method's performance and generalization ability, severely limiting the method's scalability to problem sizes (e.g., the number of points) far smaller than those specialized algorithms can handle, making the approach impractical for most real-world applications. The paper lacks related discussions and experiments.

- Following the last point, there should be experiments to study the scale of the problem sizes that the proposed framework can handle.

- There is no experiment with diffusion-based counterparts that operate directly in the parametric or symbolic solution space, which can be a helpful baseline to better understand the benefits of the pixel-space diffusion model for solving challenging geometric problems.

**Questions:**

Based on the recent study with Veo3[A], could the authors discuss the potential of using video generation models as solvers for the geometric problems investigated?

[A] Video models are zero-shot learners and reasoners.

My primary concern with the proposed method is the poor scalability despite a fancy direction for solving geometric challenges, as mentioned in the weaknesses section; thus, my current rating is slightly negative.

---

### Official Review · Reviewer_NXZd · 2025-10-31

**Soundness:** 3
**Presentation:** 3
**Contribution:** 2
**Rating:** 2
**Confidence:** 4

**Summary:**

In this work, the authors tackle solving 3 geometric problems using diffusion models. Specifically, they tackle the jordan square problem, steiner tree problem, and the maximal area polygon problem. They provide the geometric input as a condition (concatenated along with image) to the diffusion model, and the diffusion model progressively denoises to generate one possible solution. Sampling multiple times generates diverse solutions.

**Strengths:**

- I really like the setup of solving geometric problems using diffusion models. The authors demonstrate good findings about the potential of using diffusion models for solving such NP-complete / NP-hard problems.
- The paper is well-written and easy to follow. The authors describe each geometric problem well and provide adequate quantitative and qualitative results.

**Weaknesses:**

- While I really like the problem setup, the methodology is fairly weak / simple. The authors curate the dataset as a collection of solution images and train DDIM model on this dataset. There is no methodology novelty (either in terms of architecture or loss function), hence I think the paper is not fully upto ICLR's benchmarks.
- The authors listed several diffusion-based works in the Related Works section but did not provide any results comparing against them.
- Figure 7 and Figure 8 are not clear. The caption mentions left/middle/right, but there are 4 columns in the figure. So what is left/middle/right exactly?
- I appreciate the authors showing results on unseen conditions (13-15 points in MAXAP etc). At the current stage it looks like the diffusion model is still not very good at extrapolating to unseen constraints.

These are my opinions, but I will also carefully look at other reviewers' response as well as the authors' rebuttal and am open to increasing the score based on them. This is because I really like the problem setup but the methodology is just a simple application of DDIM.

**Questions:**

- I understand the authors run experiments using DDIM in image/pixel space. It would be interesting to see if latent-space diffusion models like Stable Diffusion etc could also tackle such tasks.
- Did the authors check the diversity of the output and could they quantify a diversity measure? Basically to understand how different the solutions are compared to the GT. And when sampling multiple times, what fraction tends to be a repetition?

---

### Official Review · Reviewer_4EKE · 2025-10-31

**Soundness:** 3
**Presentation:** 3
**Contribution:** 3
**Rating:** 4
**Confidence:** 5

**Summary:**

This work introduces a novel paradigm for solving complex geometric problems by reformulating them as image-to-image generation tasks. The authors demonstrate that a standard visual diffusion model, operating purely in pixel space, can be trained to denoise a random signal into an image representing a valid, approximate solution. This "geometric solver" approach is successfully applied to three classic 2D problems: the Inscribed Square Problem, the Steiner Tree Problem, and the Maximum Area Polygon Problem. While all experiments are in 2D, the work presents a new conceptual tool that could be highly relevant for 3D geometric challenges.

**Strengths:**

1. Generality and Simplicity: The method's primary strength is its generality. A single, standard U-Net architecture is used across three different, notoriously hard problems without requiring specialized, domain-specific modules or complex symbolic representations.

2. Novel Formulation: The core idea of treating hard geometric constraints as a denoising process in image space is highly novel. This bypasses traditional solvers and graph-based representations, recasting geometric reasoning as a learned generative task.

3. Strong Empirical Results (in 2D): For the 2D tasks, the model produces solutions that are surprisingly high-quality and very close to optimal (e.g., an average length ratio of 1.0008 to the optimal Steiner tree). The model also shows some ability to generalize to more complex instances than seen during training.

**Weaknesses:**

1. No Extension to 3D: The most significant weakness, especially in the context of 3D vision, is the complete lack of 3D experimentation or discussion. The paper's claims of generality are unproven, as it's unclear how this pixel-space approach would scale to 3D representations. Obvious 3D extensions (e.g., 3D Steiner trees for skeletonization, maximal-volume simple polyhedra, or surface reconstruction from points or multi-view images) are never mentioned.

2. Scalability of Representation: The reliance on 2D rasterization is a critical limitation. This approach scales quadratically with resolution ($N^2$) and introduces discretization errors (requiring a "snapping" post-processing step). A naive extension to 3D (e.g., voxels) would scale cubically ($N^3$) and be computationally prohibitive.

3. Fragility on Harder Instances: The method's robustness is questionable as problem complexity increases. The "Valid Trees Rate" for the Steiner Tree problem (Table 2) plummets from 0.996 (for 10-20 points) to 0.334 (for 41-50 points). Similarly, the "Valid Polygons Rate" (Table 3) drops from 0.953 to 0.620 when moving from 7-12 points to 13-15 points. This suggests the approach may fail on real-world, large-scale (3D) geometric problems.

4. Reliance on Post-processing: The method is not truly end-to-end. The model's image output must be converted back into a valid geometric structure using problem-specific heuristics (e.g., node detection, edge extraction, graph validation, and rigid snapping). This reliance on hand-crafted post-processing undermines the claim of a single, general framework.

**Questions:**

Please refer to the weakness part.

---

### Note · Authors · 2025-11-14

I have read and agree with the venue's withdrawal policy on behalf of myself and my co-authors.